# Diagnosis of Protected Agriculture in Imbabura—Ecuador, Period 2016–2023

**Luis Marcelo Albuja-Illescas** [1,*] **, Andrés Manolo Carrión-Burgos** [1] **, Rafael Jiménez-Lao** [2] **and María Teresa Lao** [3]

1. Agrobiodiversity and Food Security Research Group-GIASSA, Agricultural and Environmental Science Faculty, Universidad Técnica del Norte, Av. 17 de Julio 5-21 y Gral. José María Córdova, Ibarra 100105, Ecuador; amcarrion@utn.edu.ec
2. Department of Engineering, University of Almería, La Cañada de San Urbano s/n, 04120 Almería, Spain; rjlao717@gmail.com
3. Agronomy Department, Research Center for Mediterranean Intensive Agrosystems and Agrifood Biotechnology CIAMBITAL, Agrifood Campus of International Excellence ceiA3, University of Almería, 04120 Almería, Spain; mtlao@ual.es
* Correspondence: lmalbuja@utn.edu.ec; Tel.: +593-98-793-8737

**Abstract:** Protected agriculture in Ecuador began in the 1990s and has expanded due to its comparative advantages over open field production. However, there are no statistics on this sector, which limits decision-making. The aim of this research was to provide a baseline of greenhouse agriculture in Imbabura. Sentinel-2 satellite imagery was used to estimate the spatial distribution of plastic-covered surface area in 2016 and 2023. To minimize biases in estimation, manual verification was also conducted. Based on population data, a structured survey was administered to a probabilistic sample of 234 greenhouses. The results highlight the presence of 1958 greenhouses that cover 527 hectares, with an average of 0.26 hectares. The greenhouses were characterized in terms of their design, construction materials and equipment. The main crop under plastic is tomato, with 76.9%, of which the management characteristics and the productive and economic results obtained in 2023 were identified. The findings could inform the formulation of public policies or specific interventions to strengthen protected agriculture in the region; however, support mechanisms are needed to fully exploit its potential. Among these, producer organization could be a viable strategy to address food security challenges in the context of climate change.

**Keywords:** greenhouses; tomato; agricultural baseline; Sentinel-2; spatial distribution; greenhouse geodatabase

## 1. Introduction

Protected agriculture is agricultural production that uses protective structures such as greenhouses and shade houses to protect crops from climatic and biological damage to improve growing conditions. Compared with open field production, it has many advantages, including higher yield, better product quality and better access to the export market [1–3].

The greenhouse area around the world is mainly in China (82,000 ha), Spain (70,000 ha), Korea (51,787 ha), Italy (42,800 ha), Turkey (42,384 ha), Morocco (20,000 ha) and Mexico (20,000 ha), among others [4]. Worldwide, the total hectares protected with greenhouses and large tunnels with plastic was 687,350 hectares in 2004 and is estimated to have grown to 5.2 million hectares by 2013 [5,6]; with China standing out for showing the greatest growth in recent years [7].

In 2023, the Red Agricola publication on the current state of protected crops and mulching worldwide, mentions that we currently stand at approximately 5.6 million hectares [8].

Ecuador is a country with an agricultural vocation where the agricultural sector contributes nearly 8% to real gross domestic product (GDP), generates 29.4% of jobs nationwide and is a supplier of food and industrial raw materials and exportable surpluses for international exchange with capital goods [9,10].

In Ecuador since the 1960s, the use of plastic represented a revolution in the agricultural field, where the most common uses were irrigation systems, waterproofing, padding and covering [11]. The development of protected agriculture as we currently know it in Ecuador began in the early 1990s for the production of flowers, and 5 years later this system was extended to new crops such as tomato and strawberries. According to the information presented by [12], the country had approximately 1000 hectares of greenhouses, while in [13] they mention that in 2014 there were around 2700 hectares of greenhouses nationwide.

Imbabura is located in the north of Ecuador; according to the Territorial Development and Planning Plan, 2015–2035 [14], it has a territory of 4791.32 km$^2$, made up of six cantons, 36 rural parishes and 6 urban ones. Regarding land use, the Agricultural Public Information System [15] indicates that 21.4% is used in agricultural activities, and the main crops for the planted area are soft corn, banana, sugar cane, beans, avocado, peas and cocoa, and on a smaller surface, fruits, vegetables, tubers and cereals, among others.

Protected agriculture in the province of Imbabura is growing; however, there are no statistics on the sector and no research, and the productive, technical and economic situation it obtains is unknown, scenarios that could limit its development and affect decision-making at the public and private level. Villagrán et al. [16] point out that ignorance of the technical conditions of greenhouses can generate productive limitations in yield, quality and health of the final harvested products. Gathering the necessary data to analyze the current situation of greenhouses in the region would allow us to have real information on the dominant typology in each area, which would make it possible to obtain new proposals for the growth of the sector [17].

Based on this context, the objective of this work was to build a baseline of the protected agriculture sector in Imbabura and analyze its growth between the years 2016 and 2023, with emphasis on the characteristics of the greenhouses, the main crops that are managed and the productive and economic results that reach producers as initial input for the generation of research and decision-making.

Several authors emphasize the global growth of protected agriculture, a trend that is also evident in Imbabura. However, due to the lack of data and statistics on the sector, its growth cannot be quantified, nor can public policies or private interventions be designed to support these farmers.

Likewise, the availability of an updated cartography of greenhouses dedicated to horticultural crops constitutes a powerful tool to develop subsequent studies and public and private planning, and is therefore of great interest given the importance of the sector for agriculture in the province of Imbabura in the context of climate change [18].

Wu et al. [1], in their study, shows that having a baseline for protected agriculture in Mexico has allowed the implementation of public and private projects to strengthen this sector, benefiting the economy of both the producers and the state.

At an international level, various studies have focused on analyzing the dynamics of the protected agriculture sector in their respective regions as a basis for public and private decision-making. Protected agriculture has been studied as an adaptation measure to climate change for food security [19,20]. It is also addresses in terms of how this type of agriculture can adapt to extreme environments [21], as well as in description of

protected agriculture at the territorial level in Kuwait [22], Turpan [23], and Australia [24]. Additionally, Kumar et al. [25] conducted an economic analysis of protected agriculture.

Additionally, other authors, address the challenges faced by protected agriculture, along with the development of the technologies required for greenhouse-based agriculture [26–30].

## 2. Materials and Methods

### 2.1. Description of the Study Area

The study was carried out in the province of Imbabura in the year 2023. The province is located in the inter-Andean or mountain region, north of Ecuador, between the coordinates $0°21'–0°72'$ N latitude and $77°48'–79°12'$ W longitude and occupies an area of 479,489.18 ha; its altitudinal range goes from 160 to 4939 m.a.s.l. It is distributed in six cantons: Antonio Ante, Cotacachi, Ibarra, Otavalo, Pimampiro and Urcuquí [14].

### 2.2. Estimation of the Productive Surface in Greenhouses

To estimate the productive surface under plastic, the methodology utilized by [31–33] was used, which includes the use of spectral indices extracted from high-resolution satellite images; the analyzed spectral bands, band width and resolution are presented in Appendix A.

For this campaign, we have chosen to use images from the Sentinel 2 satellite, which are freely available. The Sentinel 2A scenes are part of the Copernicus Sentinel data for the years 2015–2016. For the year 2023, high-resolution Planet NICFI satellite images were used (20 m general resolution and 10 m in the visible spectrum). Google Earth Engine and ArcGis 10.8 software were used for processing.

The shortwave infrared (SWIR) bands facilitate the initial identification of greenhouses (Figure 1).

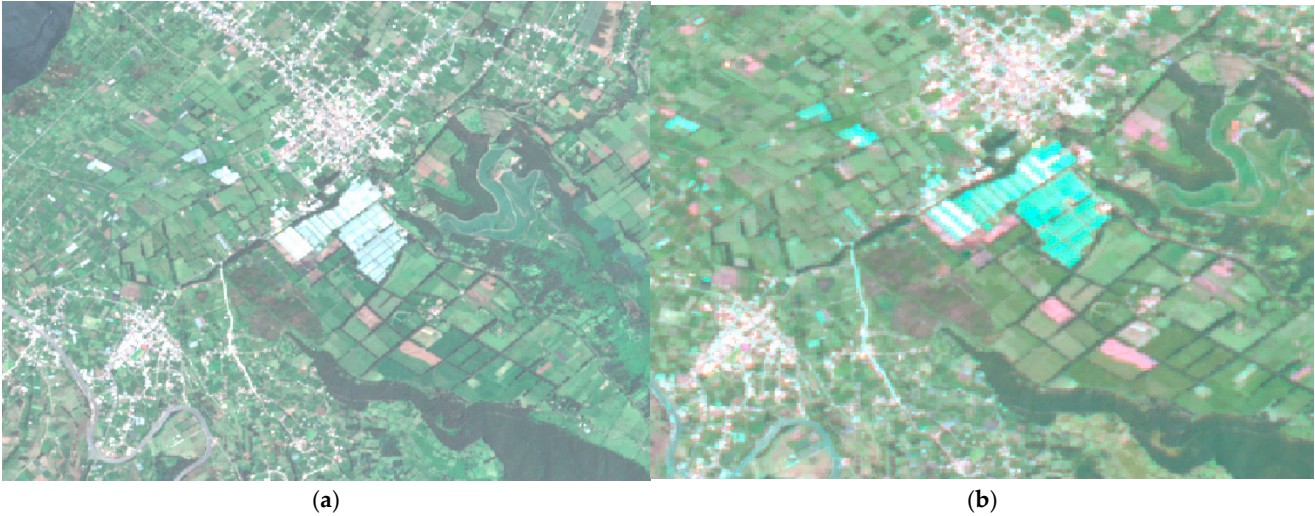

(**a**)                                                    (**b**)

**Figure 1.** Sentinel 2A image in two band combinations: (**a**) Shortwave infrared (2190 nm)–red-blue; (**b**) true color. The shortwave infrared (2190 nm) and blue bands facilitate the discrimination of greenhouses.

Protected crops have a spectral response in the visible, near-infrared and mid-infrared regions, which allows them to be differentiated from other surfaces, although these differences between the response of a greenhouse and other surfaces are not always simple.

This is why the use of specific indices and sample areas is required to help us make a correct detection. It is worth highlighting the use of the Plastic Greenhouse Detection Index (PGHI), which is shown in Equation (1) [34] (Figure 2).

$$PGHI = P\ blue/P\ SWIR2 \tag{1}$$

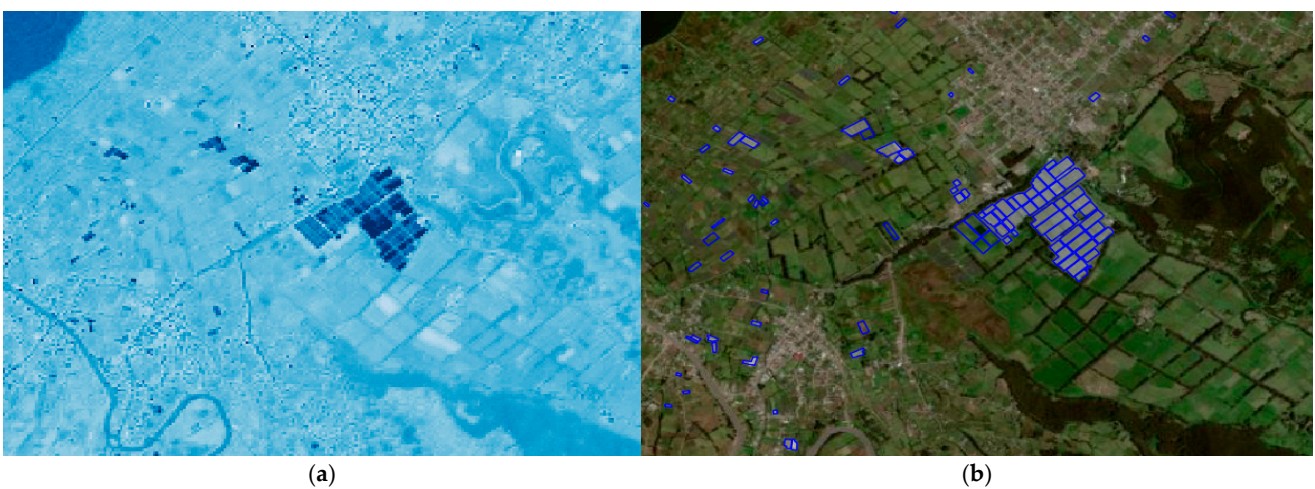

(**a**)                                    (**b**)

**Figure 2.** (**a**) Detail of the PGHI index calculated through a Sentinel 2A image. (**b**) Definition of greenhouse boundaries in high-resolution images.

*2.3. Construction Characteristics of the Greenhouses and Technical, Productive and Economic Aspects of the Tomato (Solanum lycopersicum L.)*

For its part, to identify the characteristics of the greenhouses and the tomato, 234 surveys were applied to a stratified probabilistic sample based on the total greenhouse population determined in this study. The surveys were applied between the months of June 2023 and January 2024.

The survey was made up of 7 sections with owner data, general information about the farm, structure and equipment of the greenhouses, crop management characteristics, production costs of the campaign, destination of the production and the endogenous characteristics of the producer. The questionnaire proposed by [35] was taken as a questionnaire reference.

Likewise, the economic results of tomato production were determined in the year 2023 for 1 ha; the calculated parameters were as follows: production cost (Equation (2)), total income (Equation (3)), gross profit (Equation (4)) and the cost–benefit relationship (Equation (5)). These are the same ones that were processed in the Infostat software 2020e [36]. Further information is provided in Appendix B.

$$\begin{aligned}
\text{Total Production Cost} = \Sigma\ (\text{labor cycle} + \text{Supplies} * \text{Price} + \text{Material} * \text{Price} \\
+ \text{Land rent} + \text{Marketing} + \text{Depreciation} + \text{Capital interest} + \text{Miscellaneous})
\end{aligned} \tag{2}$$

$$\text{Total income} = \text{Production} * \text{Prince} \tag{3}$$

In the case of tomato, the gross profit is equal to the total income minus the total costs of the production cycle and was calculated using Equation (4).

$$INb = It - Ct \tag{4}$$

where INb represents the gross profit, It is the total income, which was determined based on the production obtained multiplied by the mean price of the crop from the last year and Ct they are the total costs, which include land preparation, labor costs, inputs as direct

costs and the cost of money, depreciation of infrastructure and equipment and land rental as indirect costs.

The cost–benefit ratio is the relationship between the income obtained and the total production costs, as expressed in Equation (5) [37,38].

$$C/B = It/Ct \qquad (5)$$

For the collection of economic data in the field, producers were asked about the last production cycle of kidney tomatoes they had or were cultivating at the time of the survey, assuming they had the available information.

## 3. Results

### 3.1. Spatial Distribution of the Productive Surface Under Plastic in Imbabura

Figure 3 shows the spatial distribution of the productive surface under plastic in the province of Imbabura. The polygons are mapped and georeferenced for the years 2016 and 2023; the individual surface of the plastic covers is available and the increase in number was analyzed.

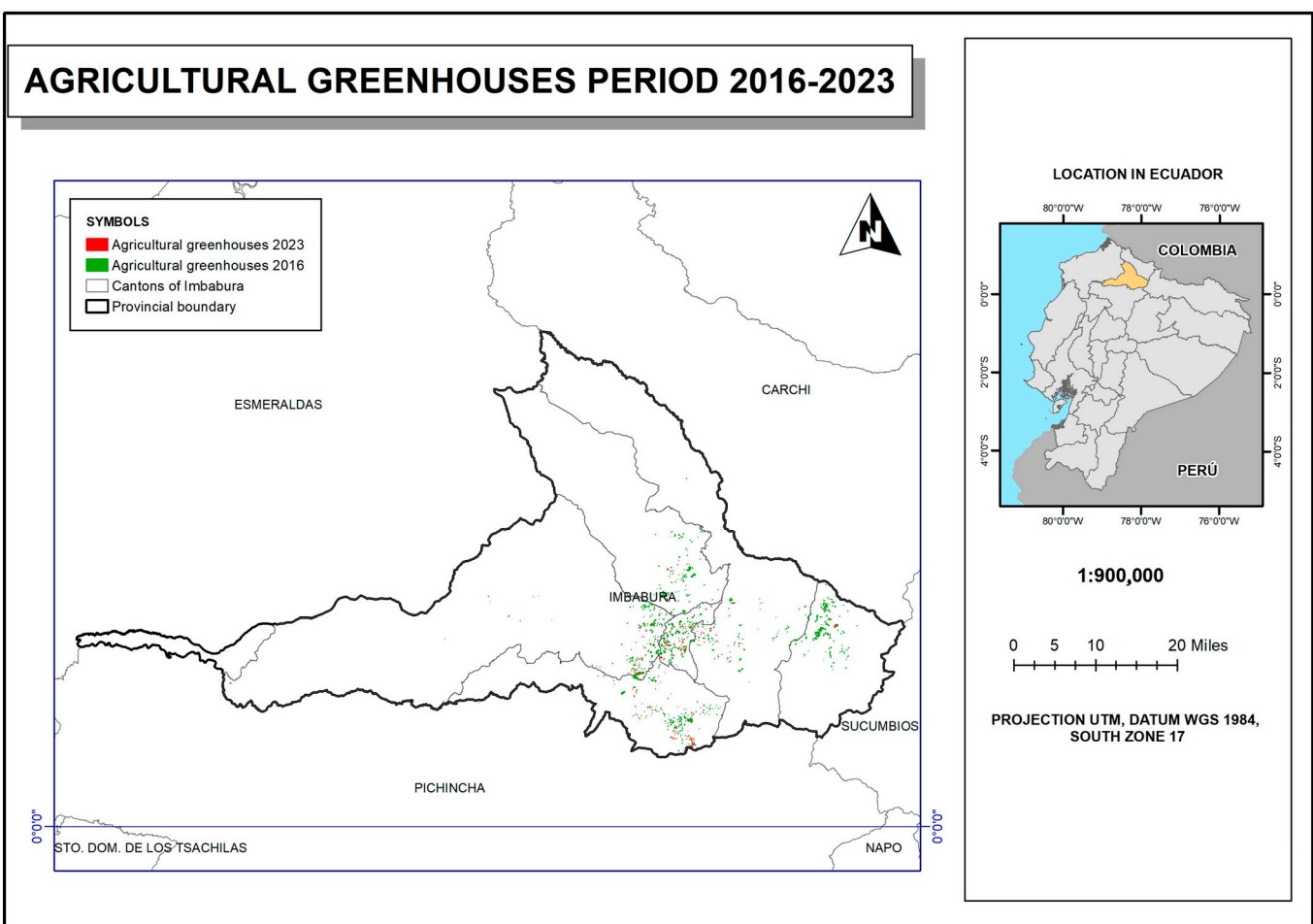

**Figure 3.** Spatial distribution of greenhouses in Imbabura in the years 2016 and 2023.

In 2016, there was an area under plastic cover of 482.42 ha in 1643 greenhouses, and by 2023 an increase of 44.61 ha is evident, which generates a total of 527.03 ha in 1958 greenhouses. The average surface area of the greenhouses is 0.269 ha, with a minimum surface area of 0.02 ha and a maximum surface area of 4.07 ha (S.D. 0.2433).

Song et al. [39] point out that reliable and timely information on agricultural production is essential to guarantee global food security, where satellite data offer the possibility of improving the monitoring of global agriculture, and agrees with [40] on the importance of mapping and classifying the different types of protected agriculture to understand the pattern of crop production.

However, the development of this type of agriculture must seek sustainable intensification, which also implies the conservation of existing natural areas [41]; in addition, greenhouses play an increasingly important role in the food supply [42].

Figure 4 shows that the surface covered with agricultural greenhouses is distributed in all the cantons of the province, with the greatest presence in Cotacachi, Otavalo, Antonio Ante and Pimampiro. This also demonstrates a growth in the number of greenhouses, with 18%, 24.5% and 21.1%, respectively, for the analysis period. This follows a global trend, as the sector is growing rapidly, and they expect it to increase in many agrarian countries [43].

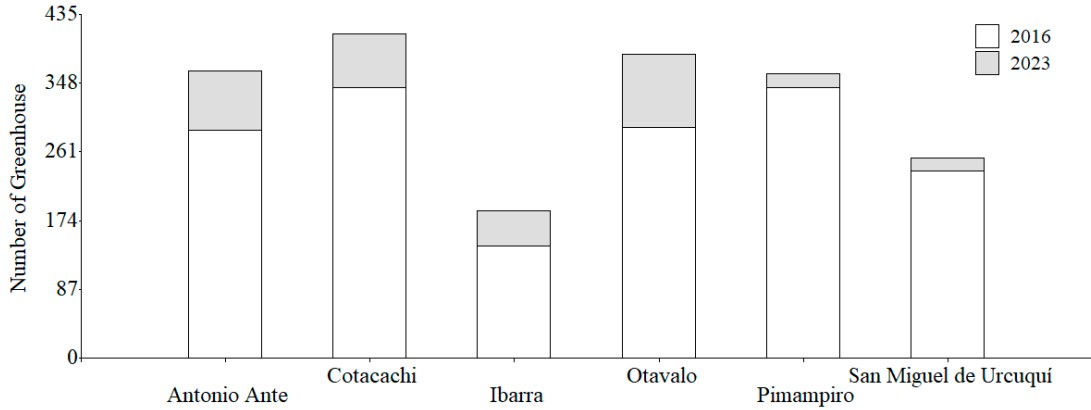

**Figure 4.** Number of greenhouses and their cantonal distribution in Imbabura by 2023.

Within the mentioned cantons, the parishes that have the highest concentration of greenhouses are Pimampiro, Atuntaqui, Cotacachi, Imantag, Quiroga, González Suárez, Tumbabiro and San Pablo. This information is important as knowledge of the location of greenhouses is essential for the implementation of territorial planning strategies [44].

Horticultural greenhouses are relatively small; as a classification, a proposal for surface ranges was made, as shown in Figure 5.

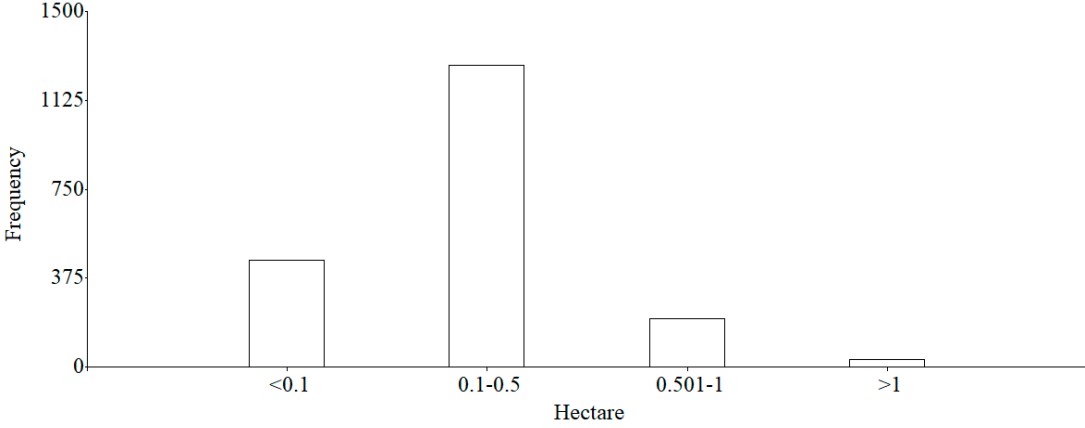

**Figure 5.** Number of greenhouses according to surface ranges.

As can be seen, 88% of greenhouses have a surface area of less than 0.5 ha, and only 1.6% of greenhouses are larger than 1 ha. The size of greenhouses is influenced by the

limited land ownership in the province; according to data from the last agricultural census, 49% of the Agricultural Production Units (UPAs) in Imbabura have less than 1 ha [45]. This situation, according to [46], is a determining factor in the vulnerability of small producers. In Mexico, large horticultural greenhouses are more than 15 hectares, and in Turkey the greenhouses have an average of 4100 m$^2$ [1,47].

Another determining factor of the size of fruit and vegetable greenhouses is the initial investment in infrastructure; the results show that, on average, in 2023 the cost per m$^2$ for a metal greenhouse was USD 12.5, for a mixed greenhouse it was USD 9 and for a wooden greenhouses it was USD 8.

### 3.2. Characteristics of the Agricultural Production Units (UPAs) That Have Fruit and Vegetable Greenhouses

The UPAs that have greenhouses have some distinctive characteristics compared to the common farms in Imbabura; these criteria are shown in Table 1.

**Table 1.** General characteristics of the UPAs that have fruit and vegetable greenhouses.

| Characteristics | Parameters | | |
|---|---|---|---|
| Land ownership (%) | Own | Leased | At the beginning |
| | 68.8 | 24.8 | 6.4 |
| Frequency of water availability (%) | Diary | Weekly | Biweekly |
| | 32.9 | 43.6 | 23.5 |
| No. greenhouses per farm | Minimum | Maximum | Media |
| | 1 | 10 | 2.3 |
| Reservoir size in m$^3$ (13% do not have) | 40 | 90 | 400 |
| Access roads to the farm (%) | Adequate | Not very suitable | |
| | 91 | 9 | |
| Access to basic services | Has | Does not have | |
| Waste collection (%) | 73.9 | 26.1 | |
| Electric light (%) | 86.8 | 13.2 | |
| Sewerage (%) | 65 | 35 | |
| Drinking water (%) | 84.6 | 15.4 | |

The farms that have greenhouses have availability of water and reservoirs, unlike the common farms that do not have them, and they are also located in locations with greater access to basic services, factors that influence production and improve marketing conditions.

In addition, these farms have greater capital assets, which enables better access to productive credit, and to production factors to organize and/or plan their production based on the market.

### 3.3. Typology of Horticultural Greenhouses in the Province of Imbabura

The types of horticultural greenhouses have a significant amount of variability in terms of surface, dimensions, location, materials and name. These are built empirically, and as a classification. the main characteristics of two types of traditional greenhouses are presented in Table 2.

**Table 2.** Typology of traditional fruit and vegetable greenhouses in Imbabura.

| Description | Type: Wood |
|---|---|
| The support structure is made of wood, eucalyptus is commonly used (*Eucalyptus globulus Labill*), with cemented posts. The roof is flat asymmetrical, with a zenithal opening, long-lasting polyethylene plastic cover and ventilation areas on all four sides.<br><br>Number of sections (n): 12 front and 8 rear<br>Section width (m): 5.85<br>Total width (m): 70.2 front and 46.8 post<br>Greenhouse length (m): 84<br>Covered area (m²): 4914<br>Zenith ventilation area (m²): 268.8<br>Side ventilation area (m²): 302.4<br>Front ventilation area (m²): 252.8<br>Total ventilation surface (m²): 824<br>Ventilation ratio (Stv/Sc, %): 16.7<br>%Minimum height (m): 4.23<br>Maximum height (m): 5.15<br><br>Some authors can reference it with the following:<br>- Chapel<br>- Modified gable<br>- Asymmetrical plane | |

| Description | Type: Metal |
|---|---|
| The entire structure is metal, with cemented posts. The roof is curved, with a zenith opening covered in long-lasting polyethylene plastic and ventilation areas on all four sides.<br><br>Number of sections (n): 3<br>Section width (m): 5<br>Total width (m): 15<br>Greenhouse length (m): 20<br>Covered area (m²): 300<br>Zenith ventilation area (m²): 24<br>Side ventilation area (m²): 42<br>Front ventilation area (m²): 0<br>Total ventilation surface (m²): 66<br>Ventilation ratio (Stv/Sc, %): 22<br>%Minimum height (m): 4<br>Maximum height (m): 5<br><br>Some authors can reference it with the following:<br>- Multi-tunnel<br>- Curved<br>- Lowered arch | |

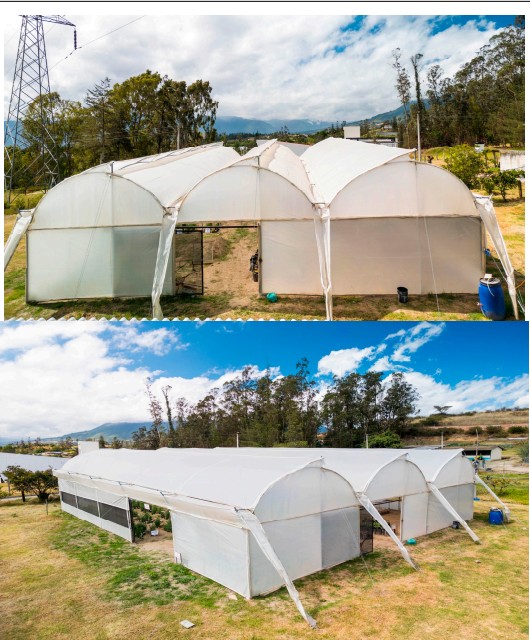

In Imbabura, greenhouses are known by common names based on the material of the structure, whether wood, metal or mixed (wooden columns and metal roof), and not the construction design. In this research, to reference the types of greenhouses we refer to the following authors: [16,48–50]. Greenhouses with a low technological level are typically found in Latin American and Asian countries, unlike high-tech greenhouses in northern European countries [51].

The cover plastic generally used is commercial polyethylene, with a useful life of 3 to 4 years; 95.3% of producers do not handle it until they change it and are unaware of the influence of plastic films on the microclimatic conditions inside the greenhouse [52].

Likewise, the dimensions of the greenhouses vary in surface area and height; at a general level, the relationship between the number of sections they have and the required climate control is not considered [53]. Generally, they are greenhouses with low-cost materials, with passive ventilation systems, and no technical considerations are made for their design and implementation, a situation similar to what was found in the neighboring country Colombia [54–56].

*3.4. Descriptive Characteristics of Fruit and Vegetable Greenhouses in Imbabura*

Table 3 shows the types of greenhouses in Imbabura, some construction characteristics, materials and the availability of equipment.

**Table 3.** Descriptive characteristics of the greenhouses in Imbabura.

| CRITERION | Parameters | | |
|---|---|---|---|
| Greenhouse type (%) | Mixed 57.1 | Wood 19 | Metal 23.9 |
| Average construction cost (USD/m$^2$) | 12.5 | 8 | 9 |
| Greenhouse years | Minimum <1 | Maximum 30 | Mean 5.3 |
| Plastic type (%) | Polyethylene without additives (PE) 27.8 | Low density polyethylene (LDPE) 5.6 | Long-lasting polyethylene (PEld) 66.6 |
| Ventilation (%) | Natural 100 | Artificial 0 | |
| Irrigation system (%) | Aspersion 0.8 | Drip 99.2 | Nebulization 0 |
| Insect mesh (%) | Has 42.3 | Does not have 57.7 | |
| Shading mesh (%) | 2.1 | 97.9 | |
| Fans (%) | 0 | 100 | |
| Nebulizers (%) | 0 | 100 | |
| Heaters (%) | 0 | 100 | |
| Temperature sensor (%) | 3.8 | 96.2 | |
| Humidity sensor (%) | 3.8 | 96.2 | |
| Collects rainwater (%) | Makes 28.2 | Not done 71.8 | |
| Greenhouse whitewashing (%) | 4.7 | 95.3 | |

In Imbabura, the greenhouses are 5.3 years old (on average), which shows the growth of the sector in recent years. However, almost all greenhouses do not have additional equipment or instruments for crop management, cultural work is based on the producer's experience and there is no external or internal climate monitoring [57].

Faced with this situation, Katzin et al. [58] show that advances in greenhouse technology seek to adjust internal climate factors, such as temperature, humidity, light intensity and $CO_2$ concentration, as these play a vital role in achieving high crop production, it is identified that most studies focus on the climate control of greenhouses [59].

The adoption of innovative technologies by producers is determined, among other factors, by testing capacity and observation, along with their personal and financial situation [60].

Furthermore, 57.7% of producers do not have insect nets, nor do they use other control methods. In this sense, Rustia et al. [61] suggest that integrated pest management is one of the central components of an effective pest control program, along with efficient greenhouse management.

Regarding irrigation, 99.2% of producers use drip systems, which is a characteristic of horticultural greenhouses in temperate or warm temperate zones in developing countries [62]. Likewise, 71.8% of producers do not collect rainwater, thereby losing enormous potential to cover irrigation demand through collection tanks or ponds [63].

### 3.5. Main Crops Present in Fruit and Vegetable Greenhouses

The main crops produced in the greenhouse are as follows: tomato (*Solanum lycopersicum* L.), which represents 76.9%; pepper (*Capsicum annum* L.), with 13.7%; pickle (*Cucumis sativus* L.), with 5.6% and, more recently, blueberry, strawberries, blackberry, grapes and celery, with the remaining percentage. These are crops that, in open fields, have phytosanitary problems and high economic risk. A similar distribution of vegetables and fruits was found in Mexico, and it is also mentioned that products obtained in greenhouses have better prices on the market than those produced in the open fields [1].

The production of tomato in a greenhouse emerged as a productive and economic alternative to traditional crops in Imbabura; where the Sierra region is the main tomato production area, and the provinces with the highest production in 2020 were Imbabura, Manabí and Pichincha, with 971 ha, 579 ha and 263 ha, respectively [64].

The producers rotate crops in the greenhouse; however, they maintain the tomato as their main crop, which is why crop management characteristics under these conditions are presented below.

### 3.6. Management of Tomato Under Plastic in Imbabura

Table 4 shows the management characteristics carried out by producers when growing tomatoes in Imbabura under greenhouse conditions, information that allows the identification of elements that can influence the productive results obtained. The table shows the differences with open field productive systems.

It can be highlighted that there are certain producers who carry out soil and/or foliar analysis of their crops; however, the use of agrochemicals predominates in both fertilization and management of pests and diseases, which accentuates major environmental problems [65] and leads to the contamination of aquifers by nitrates [66]. The use of agrochemicals in greenhouses is even greater in number and quantity than in open fields, as investigated by [67].

It total, 97.4% of producers carry out manual transplanting of seedlings; Liu et al. [68] indicate that this activity has high labor intensity, poor consistency of work quality and low efficiency.

Regarding the microclimate of the greenhouses, the producers do not carry out any monitoring, and their cultural tasks, such as irrigation, fertilization and ventilation, among others, are carried out empirically. Some producers, as a daily practice, open curtains early in the morning and close them at the end of the work day. Gong et al. [69] show that an adequate ventilation and water management strategy is beneficial to increasing the yield and quality of greenhouse crops.

Regarding post-harvest practices, producers do carry out some basic tasks, such as classification and packaging for delivery to both the wholesale market and to supermarkets, following the client's technical sheets.

**Table 4.** Management characteristics of the tomato crop in the greenhouse.

| CRITERION | Parameters | | |
|---|---|---|---|
| | Makes | Not done | |
| Soil analysis (%) | 43.6 | 56.4 | |
| Leaf analysis (%) | 17.5 | 82.5 | |
| Water analysis (%) | 21.8 | 78.2 | |
| Renewal pruning (%) | 31.6 | 68.4 | |
| Soil disinfection (%) | Chemical | Biological | Not done |
| | 75.2 | 8.1 | 16.7 |
| | Tractor | Manual | |
| Land preparation (%) | 80.3 | 19.7 | |
| | Organic | Chemical | |
| Soil fertilization (%) | 19.7 | 80.3 | |
| | Direct (seed) | Own seedlings | Acquired seedlings |
| Planting method (%) | 2.6 | 5.1 | 92.3 |
| Orientation of crop lines (%) | Longitudinal | Transversal | |
| | 46.6 | 53.4 | |
| | Substratum | Floor_bed | Soil_furrow |
| Production means (%) | 6.4 | 59.4 | 34.2 |
| No. Permanent workers (%) | Minimum | Maximum | Mean |
| | 1 | 20 | 2.91 |
| No. Casual workers (%) | 0 | 20 | 4.51 |
| | Biological only | just chemical | Mixed |
| Disease Control (%) | 1.3 | 74.4 | 24.3 |
| Pest control (%) | 1.3 | 76.9 | 21.8 |
| Monitor pests and diseases (%) | And | No | |
| | 83.3 | 16.7 | |
| | Fertirriego only | Foliar only | Mixed |
| Method of fertilizing (%) | 67.1 | 6.8 | 26.1 |
| | Bioinsumos | Chemical | Mixed |
| Inputs to fertilize (%) | 3.8 | 38.5 | 57.7 |
| | Automated | Manual | |
| Irrigation (%) | 27.8 | 72.2 | |
| | Manual | Chemical | |
| Weed Control (%) | 65 | 35 | |
| | Bees | Natural (air) | |
| Pollination (%) | 6.4 | 93.6 | |

### 3.7. Productive and Economic Indicators of Tomato

The productive and economic results of tomato production are shown in Table 5.

**Table 5.** Productive and economic indicators of tomato mean value in 2023.

| Variable | Mean | S.D |
|---|---|---|
| Yield (kg·m$^{-2}$) | 12.07 | 4.07 |
| Total cost (USD·ha$^{-1}$) | 47,249.46 | 1685.88 |
| Income (USD·ha$^{-1}$) | 60,372.45 | 2037.58 |
| Gross profit (USD·ha$^{-1}$) | 13,122.99 | 1806.86 |
| C/B ratio | 1.6 | 0.86 |

The average yields of the 2023 campaign are lower than those obtained by [48], also in comparison with countries like Mexico, with average yields of 18.22 kg·m$^{-2}$; Uruguay, with ranges between 6.2 and 23.1 kg·m$^{-2}$; Spain, with 28 kg·m$^{-2}$ and the Netherlands, with 60 kg·m$^{-2}$ [1,49,70,71], a situation that shows the need for new strategies to improve production.

Authors such as [72–74] mention the importance of substantially growing food production to meet future food needs, where the production of crops in greenhouses will guarantee the supply of food for the world population in the coming decades [75].

Compared to open field crops, greenhouse production presents better economic results due to environmental conditions such as light, temperature and humidity, which are all obtained from these protective structures [76].

However, the results under the greenhouse are higher than those mentioned by the Continuous Agricultural Surface and Production Survey [77], in its economic module, where it estimates that the average income for areas up to 1 hectare is USD 9739, average operational costs is USD 7567 and the average profitability is USD 1573.

The marketing destinations are focused on the wholesale markets of the city of Ibarra (Imbabura) and Quito (Pichincha), and in the country's supermarkets, the presentations vary and comprise 20 kg drawers, 18 kg cartons and 1 kg covers. Daza et al. [78] report that short marketing circuits have been promoted at the national level to shorten the commercial chain so that producers can obtain better prices.

In total, 32% of producers are not satisfied with the price of tomatoes and point to price variability as a problem. Figure 6 shows the dynamics of the price per kg in the year 2023 in the Ibarra wholesale market.

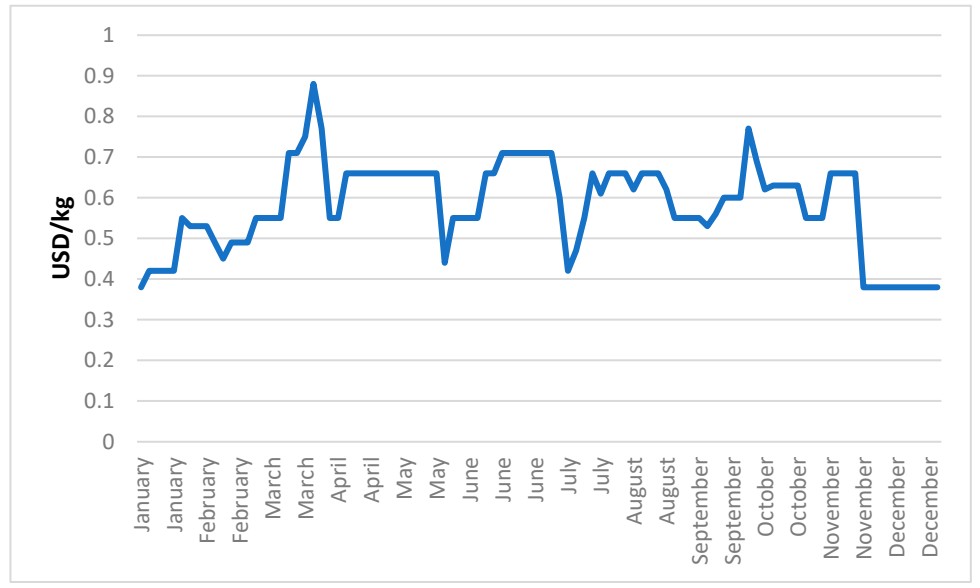

**Figure 6.** Dynamics of the price of tomatoes (kg) in the Ibarra wholesale market in 2023.

Neimark et al. [79] conclude that sustainability initiatives that work with small producers should not simply focus on high prices, as this is not enough to promote social, economic and environmental sustainability in a raw materials supply chain.

### 3.8. Endogenous Characteristics of Protected Agriculture Producers

Producers who have and manage greenhouse crops have certain particularities compared with open field producers, such as the level of education, where 17.9% of producers completed higher education, 38.5% completed secondary education and only 1.7% had no education. This is a factor that differentiates them from the profile of producers in Ecuador, who commonly have no or limited access to education. In 2022, 55.2% stated that they had finished primary school [80].

Producers in protected agriculture in Imbabura have an average age of 44.7 years. According to [81], in Ecuador, 45.9% of producers have an age that ranges between 45 and 64 years.

Furthermore, they are not organized, since only 9.4% are part of a productive and/or commercial association; they have an average of 8.7 years of experience in greenhouse cultivation and 58% are natives of the area. Likewise, 35.4% have another complementary economic occupation; which leads to an increasing proportion of farmers' income comes from non-agricultural activities [82].

Additionally, it is evident that producers who have greenhouses have greater access to training and technical advice. Figure 7 shows the availability of services that producers had access to as factors supporting production in the 2023 campaign.

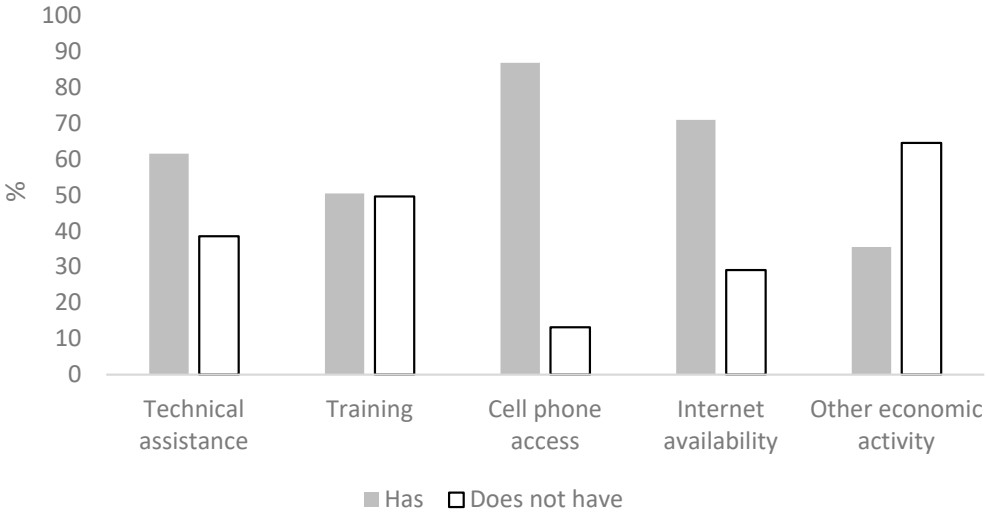

**Figure 7.** Availability of support services for greenhouse production in 2023.

For their part, Vargas-Canales et al. [83] showed the positive influence of a regional innovation system for the protected agriculture sector in Mexico, with the coordination of the State, universities and the extension service in the dissemination of new knowledge.

## 4. Discussion

The research results provide descriptive and quantitative baseline information on the situational status of protected agriculture in Imbabura. The data reveal growth in the sector, both in the number of greenhouses and the covered area, aligning with global trends [6]. A mapping of agricultural greenhouses in Imbabura is available, which will facilitate public and private decision-making and will also benefit further studies, which can use this baseline as a reference [1,22].

Protected agriculture in Imbabura, as in the neighboring country of Colombia, has a low or negligible technological level, suggesting underutilization of greenhouses. Nevertheless, it offers significant advantages compared to open-field agriculture. Similarly, the construction characteristics of greenhouses do not vary by crop type or location, meaning that agro-climatic conditions are not considered during greenhouse construction. This may be limiting to productive development.

The primary greenhouse crop in Imbabura is tomate riñón (*Solanum lycopersicum* L.), which is economically and commercially attractive. It has enabled producers to achieve yields that provide economic benefits and establish commercial agreements with private companies.

Similarly, Wu et al. [1] provide a description of protected agriculture in México and aligns with this study regarding the rapid growth of the sector and the importance of vegetables for food security and the generation of economic benefits for producers.

The results align with [16], which was a study conducted in Colombia on the characterization of greenhouse types and included a microclimate evaluation of them, where

there were several similarities regarding greenhouse architecture, the low or nonexistent technological level and the technical limitations of the producers.

Similar coincidences were found in the findings of [17] in the provinces of Argentina when the greenhouses were characterized. The similarities included the limited ventilations surface of the greenhouses, the materials used in their construction and the consequences for the crops.

The expansion of agriculture and the regulation of climate change are two critical and highly interconnected global issues [84,85]. In fact, the projected trend of unstable climatic conditions poses a threat to food security and may drive agricultural expansion and deforestation [86].

Greenhouse farming is one of the sustainable approaches to smart agriculture and is regarded as an alternative method to address the food crisis caused by rapid population growth, climate change and environmental pollution [6,87].

Agricultural greenhouses play a vital role in food production, resource conservation and rural economies. Over the past decade, a positive trend in the growth of protected agriculture has been observed globally [7].

Greenhouses can provide optimal growing conditions and ensure consistent and stable crop production. Moreover, they can mitigate the adverse impacts of extreme environmental conditions, such as dramatic temperature fluctuations, variations in light or other severe weather events, like excessive or insufficient precipitation.

## 5. Conclusions

The protected agriculture sector is clearly visible in Imbabura; the number of greenhouses and the surface under plastic both evidence this, with an increase in greenhouses of 19.17% in the study period and with prospects for constant growth.

The fruit and vegetable greenhouses in Imbabura have particular characteristics, such as a relatively small productive surface, no additional equipment, no climate monitoring is carried out and their management is totally empirical. However, the climate protection of the greenhouses and the availability of water and irrigation systems have allowed producers to achieve yields that generate economic benefits; in addition, they have been able to generate commercial ties with private companies for direct marketing.

Protected agriculture in Imbabura could be a strategy to face the challenges of a growing demand for food in the context of changing and extreme climatic conditions; however, this sector requires greater research and support to solve the problems it faces at a technical, productive and commercial level.

Among various strategies to strengthen the protected agriculture sector in Imbabura, it is recommended to implement technical training programs for producers in crop management and climate monitoring. The need for public policies that promote access to innovative technologies and financing for small producers is also highlighted, along with education in organization processes for producers, which could serve as a fundamental pillar for advancing toward a climate-resilient production system.

Several research efforts remain pending for this sector of protected agriculture, emerging from the findings achieved and the limitations identified at the productive, technological and climatic context levels. Among these, the need for studies on microclimate management, low-cost management strategies and technological advancements stands out.

**Author Contributions:** Conceptualization: L.M.A.-I. and M.T.L.; methodology, L.M.A.-I. and M.T.L.; software, L.M.A.-I. and R.J.-L.; validation, A.M.C.-B. and M.T.L.; formal analysis, M.T.L.; research, L.M.A.-I. and M.T.L.; data curation, L.M.A.-I. and R.J.-L.; writing—preparation of the original draft, L.M.A.-I. and R.J.-L.; writing—review and editing, L.M.A.-I., A.M.C.-B. and R.J.-L.; visualization, L.M.A.-I., R.J.-L. and M.T.L.; supervision, M.T.L. All authors have read and agreed to the published version of the manuscript.

**Funding:** This research was funded by the Universidad Técnica del Norte, through RESOLUTION No. UTN-CI-2024-169-R, which approves the research project titled: "DIAGNOSIS OF THE PROTECTED AGRICULTURE SECTOR IN THE PROVINCE OF IMBABURA–PHASE 1", belonging to FICAYA.

**Data Availability Statement:** The data from this research are available for both the greenhouse geodatabase for the years 2016 and 2023 and the survey database.

**Conflicts of Interest:** The authors declare no conflicts of interest.

# Appendix A

**Table A1.** Sentinel scenes are made up of 13 bands with different characteristics.

| Bands | Name | Spatial Resolution (m) | Spectral Width (nm) |
|---|---|---|---|
| B1 | Aerosol | 60 | 443 |
| B2 | Blue | 10 | 490 |
| B3 | Green | 10 | 560 |
| B4 | Red | 10 | 665 |
| B5 | Near infrared | 20 | 705 |
| B6 | Near infrared | 20 | 740 |
| B7 | Near infrared | 20 | 783 |
| B8 | Near infrared | 10 | 842 |
| B8A | Near infrared | 20 | 865 |
| B9 | Water steam | 60 | 9.945 |
| B10 | Cirrus | 60 | 1.375 |
| B11 | Mid infrared | 20 | 1.610 |
| B12 | Mid infrared | 20 | 2.190 |

# Appendix B

**Table A2.** Production cost table summary format.

| Detail | Unit | Quantity | Unit Value | Total Value |
|---|---|---|---|---|
| A. Direct Costs | | | | |
| Soil preparation | Ha | x | $x | $xx |
| Labor | Hours | x | $x | $xx |
| Supplies | Kg/ha | x | $x | $xx |
| Marketing | USD | x | $x | $xx |
| B. Indirect Costs | | | | |
| Land rent per cycle | Ha | x | $x | $xx |
| Depreciation of infrastructure/equipment | Cycle | x | $x | $xx |
| Interest on credit | % | x | $x | $xx |
| Administrative expenses | Fixed | x | $x | $xx |
| TOTAL COSTS (A + B) | | | | |

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
