# Peer review of "Diagnosis of Protected Agriculture in Imbabura—Ecuador, Period 2016–2023"

_agronomy, doi:10.3390/agronomy15010166_

Round 1

Reviewer 1 Report

Comments and Suggestions for Authors

The study addresses the important and relevant issue of the development of protected agriculture, this type of agriculture has great prospects for growth especially in the context of the growth of the world's human population and the creation of conditions for it to become resilient to climatic and environmental factors. I consider the issue important, the method used for analysis is appropriate. However, I have comments on the structure of the study. It is mandatory for the authors to introduce a literature review chapter ( after the introduction), and they should include a separate discussion of the results in the paper. You can find studies in the international literature on this issue. These two issues must be completed in order for the study to meet the requirements for publication in a journal with Impact Factor. This will make the study valuable to the international reader and more likely to be cited. 

the paper lacks two key elements that a paper submitted to a journal with Impact Factor should have, to make special remarks at the beginning you need to adjust the shape of the paper accordingly, the chapter with literature review and discussion of results is missing, the authors need to include them in order for the paper to be further analyzed by reviewers.

Author Response

The study addresses the important and relevant issue of the development of protected agriculture, this type of agriculture has great prospects for growth especially in the context of the growth of the world's human population and the creation of conditions for it to become resilient to climatic and environmental factors. I consider the issue important, the method used for analysis is appropriate. However, I have comments on the structure of the study. It is mandatory for the authors to introduce a literature review chapter ( after the introduction), and they should include a separate discussion of the results in the paper. You can find studies in the international literature on this issue. These two issues must be completed in order for the study to meet the requirements for publication in a journal with Impact Factor. This will make the study valuable to the international reader and more likely to be cited. 

the paper lacks two key elements that a paper submitted to a journal with Impact Factor should have, to make special remarks at the beginning you need to adjust the shape of the paper accordingly, the chapter with literature review and discussion of results is missing, the authors need to include them in order for the paper to be further analyzed by reviewers.

Following your advice, the literature review has been included in the manuscript after the introduction. Although there is no chapter title, as the journal's format does not present one, the writing does incorporate your recommendation, highlighted in green.

Similarly, a separate discussion chapter of the article's results has been included, highlighted in green

We have attached the revised manuscript, which incorporates your suggestions, and we remain available for any further instructions.

Reviewer 2 Report

Comments and Suggestions for Authors

    The manuscript by Albuja-Illescas et al. is devoted to general diagnostics of protected agriculture in Imbabura—Ecuador (2016–2023). The work seems to be descriptive; however, the results can potentially useful. I have some comments and questions.

   1. Introduction: Maybe, the objective of this work should be more stressed.

   2. Section 2.2: Spectral imaging from the SENTINEL 2 satellite should be described in more details. For example, what were spectral bands analyzed? What was width of these bands? What was spatial resolution of this imaging? It should be clarified because these points can be important for diagnostics of the protected agriculture.

   3. Line 104, Figure 2: What was PGHI index? What was equation to its calculate? It should be described.

   4. Figure 3: What was language used in this figure? I suppose that only English should be used.

   5. Tables 4 and 5: What does term ”media” mean?

   6. Table 5: Were these parameters calculated for all investigated period (2016-2023)?

Author Response

The manuscript by Albuja-Illescas et al. is devoted to general diagnostics of protected agriculture in Imbabura—Ecuador (2016–2023). The work seems to be descriptive; however, the results can potentially useful. I have some comments and questions.

  1. Introduction: Maybe, the objective of this work should be more stressed.

Following your advice, the manuscript has been improved in the following ways, highlighted in turquoise:

In the introduction section, the focus of the research has been emphasized and highlighted in turquoise.

  1. Section 2.2: Spectral imaging from the SENTINEL 2 satellite should be described in more details. For example, what were spectral bands analyzed? What was width of these bands? What was spatial resolution of this imaging? It should be clarified because these points can be important for diagnostics of the protected agriculture.

Following your advice, the description of the satellite imagery in section 2.2 has been expanded with information of spectral bands, band width, and resolution. Additionally, this information is presented in “Appendix A”, Table A1.

  1. Line 104, Figure 2: What was PGHI index? What was equation to its calculate? It should be described.

Following your advice, The PGHI equation used, along with the corresponding citation, has been included as Equation 1 in the manuscript.

  1. Figure 3: What was language used in this figure? I suppose that only English should be used.

Following your advice, in figure 3, the corrections have been made, and everything is only in English.

  1. Tables 4 and 5: What does term ”media” mean?

Following your advice, in tables 4 and 5, the term 'media' has been replaced with 'mean'

  1. Table 5: Were these parameters calculated for all investigated period (2016-2023)?

Following your advice, in table 5, it is clarified that the results correspond only to the year 2023.

We have attached the revised manuscript, which incorporates your suggestions, and we remain available for any further instructions.

Reviewer 3 Report

Comments and Suggestions for Authors

Dear authors

The document presents significant opportunities for improvement in its introduction, abstract, and methodology, as these sections fail to provide the necessary context, objectives, and structure to support a robust analysis. Additionally, the discussion requires greater depth to interpret the results within a broader framework and relate them to similar cases globally. It is recommended to include a chapter with practical recommendations that offers actionable solutions to the issues identified in the study, as well as to enhance the conclusions so they are more specific and oriented toward future actions. Furthermore, the attached document contains a series of specific comments that should be addressed to strengthen the quality and coherence of the work.

Regards

Author Response

The document presents significant opportunities for improvement in its introduction, abstract, and methodology, as these sections fail to provide the necessary context, objectives, and structure to support a robust analysis. Additionally, the discussion requires greater depth to interpret the results within a broader framework and relate them to similar cases globally. It is recommended to include a chapter with practical recommendations that offers actionable solutions to the issues identified in the study, as well as to enhance the conclusions so they are more specific and oriented toward future actions. Furthermore, the attached document contains a series of specific comments that should be addressed to strengthen the quality and coherence of the work.

Following your advice, the manuscript has been improved, and the changes are highlighted in fuchsia.

Following your advice, the manuscript has been improved in the introduction, summary, and methodology sections with a more detailed explanation of the context and emphasis on the objectives. Additionally, a discussion chapter has been included after the results to compare with countries in similar contexts.

Finally, specific recommendations for the protected agriculture sector in Imbabura have been included, along with an explanation of the potential public and private interventions that could be generated based on the findings of the research.

We have attached the revised manuscript, which incorporates your suggestions, and we remain available for any further instructions.

it is recommended to specify the main limitations of the study, such as possible biases in the estimation of the area or in the sample of greenhouses. In addition, it would be useful to highlight how the findings can guide public policies or specific interventions to strengthen protected agriculture in the region. Finally, more concrete suggestions could be included on the support mechanisms needed to address climate and food security challenges.

Following your advice, the manuscript has been improved, and the changes are highlighted in fuchsia.

In the abstract and methodology, the main limitations of the study were addressed, along with the efforts undertaken to minimize biases in the estimation of greenhouse areas. Additionally, it was emphasized that the results can inform the development of public policies or specific interventions in the sector. Concrete suggestions, such as support mechanisms for producers, were included and are highlighted in fuchsia in the manuscript.

These statistics are already more up to date or please modify

Following your advice, the information has been updated, and citations from articles on the growth of the protected agriculture sector worldwide have been included, with a particular emphasis on China at the global level.

it is suggested to include a clearer justification as to why it is essential to develop a baseline in this particular region. Finally, it would be valuable to highlight how the data collected could influence public policies or strategies for sustainable agricultural development in a context of climate change.

Following your advice, a clearer justification has been written on the importance of having a baseline for agricultural greenhouses and the implications it could have for the development of public policies and/or private sector interventions. This is highlighted in fuchsia.

Please include figure

In the study area description section, the map of Imbabura was not included, as a few lines below, Figure 3 presents the map, which serves the purpose of providing the geographic location and shows the location of the greenhouses present in 2016 and 2023.

I suggest including a more detailed description of the specific spectral indices used and their relevance for greenhouse identification, which would strengthen the reproducibility of the study. In addition, it would be valuable to justify the choice of SENTINEL 2 and Planet NICFI satellite images in terms of resolution, capture frequency and suitability for this type of analysis. Finally, it is recommended to include a brief discussion on the possible limitations of this methodology, such as accuracy in areas with high cloud cover or greenhouse-like structures.

Following your advice, which coincides with another reviewer, a more detailed description of the spectral bands used, the PGHI index, images resolution, and equation 1, along with Appendix A1, has been included.

Likewise, the selection of the images and their suitability for this type of analysis was made based on the work of the following authors, one of whom is part of this project.

Li, J., Zhang, Lianpeng, Shen, Y., Li, X., Liu, W., Chai, Q., Zhang, R., & Chen, D. (2020). Object-Based Mapping of Plastic Greenhouses with Scattered Distribution in Complex Land Cover Using Landsat 8 OLI Images: A Case Study in Xuzhou, China. https://doi.org/10.1007/s12524-019-01081-8

Aguilar, M. A., Jiménez-Lao, R., & Aguilar, F. J. (2021). Evaluation of Object-Based Greenhouse Mapping Using WorldView-3 VNIR and SWIR Data: A Case Study from Almería (Spain). Remote Sensing 2021, Vol. 13, Page 2133, 13(11), 2133. https://doi.org/10.3390/RS13112133

Aguilar, M. A., Jiménez-Lao, R., Ladisa, C., Aguilar, F. J., & Tarantino, E. (2022). Comparison of spectral indices extracted from Sentinel-2 images to map plastic covered greenhouses through an object-based approach. Https://Doi.Org/10.1080/15481603.2022.2071057, 59(1), 822–842. https://doi.org/10.1080/15481603.2022.2071057

It is also mentioned that to reduce estimation biases, a manual review was conducted in which the individual greenhouse polygons were plotted, which could be a potential limitation of the methodology used.

I would like to include in the conclusions specific recommendations to strengthen the protected agriculture sector in Imbabura, such as the implementation of technical training programs for producers in crop management and climate monitoring. It would also be valuable to highlight the need for public policies that promote access to innovative technologies and financing for small farmers. Finally, a broader vision could be incorporated that links the growth of the sector with the potential impact on regional food security and adaptation to climate change.

Following your advice, some specific recommendations have been included in the conclusions, such as strategies for strengthening the sector, namely: the implementation of technical training programs for producers, climate monitoring, and the organization of producers.

Additionally, the conclusions have been enriched by mentioning the potential connection of the findings with the design of public policy and/or private sector intervention in the context of climate change. All of this has been highlighted in fuchsia.

We have attached the revised manuscript, which incorporates your suggestions, and we remain available for any further instructions.

Reviewer 4 Report

Comments and Suggestions for Authors

First of all, thank you for trusting me to assess the merit of the manuscript for publication.

This is an article that has surveyed the number and area of greenhouses in the region of Imbabura, Ecuador. The authors point out that the information will be useful for planning and implementing actions to improve agriculture in the region, which is an important agricultural area in the country, according to much of the text reported in the Introduction. In fact, it could be an important tool for Ecuador.

The manuscript is very well structured and written. I made a few comments on the manuscript. 

The Introduction is pertinent in portraying the need to obtain this information. The objective is clear. The methodology is adequately detailed. The Results section provides details of the image survey, characterization of the types and other information relating to the management of greenhouses. Photographs help the text. The characterization of the structures and what is planted in the greenhouses is well done.

Author Response

First of all, thank you for trusting me to assess the merit of the manuscript for publication.

This is an article that has surveyed the number and area of greenhouses in the region of Imbabura, Ecuador. The authors point out that the information will be useful for planning and implementing actions to improve agriculture in the region, which is an important agricultural area in the country, according to much of the text reported in the Introduction. In fact, it could be an important tool for Ecuador.

The manuscript is very well structured and written. I made a few comments on the manuscript. 

The Introduction is pertinent in portraying the need to obtain this information. The objective is clear. The methodology is adequately detailed. The Results section provides details of the image survey, characterization of the types and other information relating to the management of greenhouses. Photographs help the text. The characterization of the structures and what is planted in the greenhouses is well done.

Following your advice, improvements were made to the document and are highlighted in blue.

The manuscript includes the full name of the term Gross Domestic Product, the indicated paragraphs have been sequenced, the geographic coordinates for Imbabura were completed, the scientific names were corrected to lowercase, the greenhouses from the year 2023 in Figure 3 were changed to red for better visibility, and the value 0 was changed to <1 in Table 3.

We have attached the revised manuscript, which incorporates your suggestions, and we remain available for any further instructions.

Reviewer 5 Report

Comments and Suggestions for Authors

The manuscript "Diagnosis of protected agriculture in Imbabura—Ecuador, period 2016–2023" provides valuable insights into greenhouse agriculture in the Imbabura region . The comprehensive analysis of greenhouse characteristics, management practices, and economic outcomes offers valuable insights for agricultural development. However there are gaps for improvement to enhance its impact and relevance to the scientific community. In order for the article to be considered for publication, I suggest that the authors pay attention to my following comments:

1. While the study provides valuable baseline data about greenhouse agriculture in Imbabura, the discussion section would benefit from a deeper analysis comparing your findings with other developing regions facing similar challenges in protected agriculture

2. The methodology for calculating economic indicators (costs, income, and benefit/cost ratio) needs more detailed explanation. Consider adding a supplementary table breaking down the cost components and assumptions used in your calculations

3. The paper would be strengthened by including a brief analysis of the limitations faced during the research. For instance, discuss any challenges in data collection, potential biases in the survey responses, or limitations in the satellite imagery analysis

4. The conclusions section could be more action-oriented. While you mention that the sector "requires greater research and support," consider providing specific recommendations for policymakers, extension services, and future research directions

5. Given the importance of climate change mentioned in your introduction and conclusions, the paper would benefit from a more thorough discussion of how the current greenhouse infrastructure and management practices in Imbabura might need to adapt to future climate scenarios

Good job!

Author Response

The manuscript "Diagnosis of protected agriculture in Imbabura—Ecuador, period 2016–2023" provides valuable insights into greenhouse agriculture in the Imbabura region . The comprehensive analysis of greenhouse characteristics, management practices, and economic outcomes offers valuable insights for agricultural development. However there are gaps for improvement to enhance its impact and relevance to the scientific community. In order for the article to be considered for publication, I suggest that the authors pay attention to my following comments:

  1. While the study provides valuable baseline data about greenhouse agriculture in Imbabura, the discussion section would benefit from a deeper analysis comparing your findings with other developing regions facing similar challenges in protected agriculture

Following your advice, the manuscript has been improved, and the changes are presented in gray.

In the discussion, a comparison of our findings with those from other developing regions, particularly in Latin America, has been included.

  1. The methodology for calculating economic indicators (costs, income, and benefit/cost ratio) needs more detailed explanation. Consider adding a supplementary table breaking down the cost components and assumptions used in your calculations

Following your advice, the equation 2, shows the breakdown of cost components and the assumptions used for the calculation, the equations 3 and 4, shows de income and gross profit and de equation 5 explain the cost-benefit relationship.

Appendix B has been included, which shows the questions asked to the producers in order to obtain the economic information from the last kidney tomato production cycle.

  1. The paper would be strengthened by including a brief analysis of the limitations faced during the research. For instance, discuss any challenges in data collection, potential biases in the survey responses, or limitations in the satellite imagery analysis

Following your advice, the information regarding the limitations faced during the research has been included, as well as the efforts made to reduce biases both in the application of surveys and in the use of satellite imagery.

  1. The conclusions section could be more action-oriented. While you mention that the sector "requires greater research and support," consider providing specific recommendations for policymakers, extension services, and future research directions

Following your advice, the conclusions have been enriched with specific recommendations regarding the needs for training, climate monitoring, and organization, where both public and private intervention are relevant.

  1. Given the importance of climate change mentioned in your introduction and conclusions, the paper would benefit from a more thorough discussion of how the current greenhouse infrastructure and management practices in Imbabura might need to adapt to future climate scenarios

Following your advice, it was also recommended in the conclusions that, due to climate change and the limitations of natural ventilation and equipment, research should focus on strategies that could help address this issue, possibly through new greenhouse designs and the inclusion of low-cost technology.

We have attached the revised manuscript, which incorporates your suggestions, and we remain available for any further instructions.

Round 2

Reviewer 1 Report

Comments and Suggestions for Authors

I read a well revised study, but not perfectly. Important comments were introduced in the introduction. I like the current discussion of the results. I am also pleased that the authors have developed the conclusion. A weakness is the lack of a separate chapter with a literature review. I leave this weakness to the Editor for discussion. I always stress that in papers with Impact Factors the authors should introduce a separate chapter with a review of the international literature on the topic.

Author Response

I read a well revised study, but not perfectly. Important comments were introduced in the introduction. I like the current discussion of the results. I am also pleased that the authors have developed the conclusion. A weakness is the lack of a separate chapter with a literature review. I leave this weakness to the Editor for discussion. I always stress that in papers with Impact Factors the authors should introduce a separate chapter with a review of the international literature on the topic.

Dear Reviewer,

We sincerely appreciate and value your comments and suggestions, which have greatly enhanced the manuscript.

Reviewer 2 Report

Comments and Suggestions for Authors

Authors completely considered my comments. I have nod additional remarks. 

Author Response

Authors completely considered my comments. I have nod additional remarks. 

Dear Reviewer,
We sincerely appreciate and value your comments and suggestions, which have greatly enhanced the manuscript.

Reviewer 5 Report

Comments and Suggestions for Authors

The authors have responded appropriately to my comments.

Good job!

Author Response

The authors have responded appropriately to my comments.

Good job!

Dear Reviewer,
We sincerely appreciate and value your comments and suggestions, which have greatly enhanced the manuscript.